# High Nutritional Quality of Human-Induced Pluripotent Stem Cell-Generated Proteins through an Advanced Scalable Peptide Hydrogel 3D Suspension System

**DOI:** 10.3390/foods12142713

**Published:** 2023-07-15

**Authors:** Shan Xu, Guangyan Qi, Timothy P. Durrett, Yonghui Li, Xuming Liu, Jianfa Bai, Ming-Shun Chen, Xiuzhi (Susan) Sun, Weiqun Wang

**Affiliations:** 1Department of Food Nutrition Dietetics and Health, Kansas State University, Manhattan, KS 66506, USA; shan2@ksu.edu; 2Department of Grain Science and Industry, Kansas State University, Manhattan, KS 66506, USA; guangyan@ksu.edu (G.Q.); yonghui@ksu.edu (Y.L.); 3Department of Biochemistry and Molecular Biophysics, Kansas State University, Manhattan, KS 66506, USA; tdurrett@ksu.edu; 4USDA-ARS and Department of Entomology, Kansas State University, Manhattan, KS 66506, USA; xmliu@ksu.edu (X.L.); mchen@ksu.edu (M.-S.C.); 5Kansas State Veterinary Diagnostic Laboratory, Kansas State University, Manhattan, KS 66506, USA; jbai@vet.k-state.edu

**Keywords:** hiPSCs, PGmatrix 3D suspension culture system, nutritional quality, amino acid composition, in vitro protein digestibility, regenerative agriculture

## Abstract

Cell-cultured protein technology has become increasingly attractive due to its sustainability and climate benefits. The aim of this study is to determine the nutritional quality of the human-induced pluripotent stem cell (hiPSC)-cultured proteins in an advanced 3D peptide hydrogel system for the highly efficient production of cell-cultured proteins. Our previous study demonstrated a PGmatrix peptide hydrogel for the 3D embedded culture of long-term hiPSC maintenance and expansion (PGmatrix-hiPSC (PG-3D)), which showed significantly superior pluripotency when compared with traditional 2D cell culture on Matrigel and/or Vitronectin and other existing 3D scaffolding systems such as Polyethylene glycol (PEG)-based hydrogels. In this study, we designed a PGmatrix 3D suspension (PG-3DSUSP) system from the PG-3D embedded system that allows scaling up a hiPSC 3D culture volume by 20 times (e.g., from 0.5 mL to 10 mL). The results indicated that the PG-3DSUSP was a competitive system compared to the well-established PG-3D embedded method in terms of cell growth performance and cell pluripotency. hiPSCs cultured in PG-3DSUSP consistently presented a 15–20-fold increase in growth and a 95–99% increase in viability across multiple passages with spheroids with a size range of 30–50 μm. The expression of pluripotency-related genes, including *NANOG*, *OCT4*, *hTERT*, *REX1*, and *UTF1*, in PG-3DSUSP-cultured hiPSCs was similar to or higher than that observed in a PG-3D system, suggesting continuous pluripotent maintenance. The nutritional value of the hiPSC-generated proteins from the PG-3DSUSP system was further evaluated for amino acid composition and in vitro protein digestibility. The amino acid composition of the hiPSC-generated proteins demonstrated a significantly higher essential amino acid content (39.0%) than human skeletal muscle protein (31.8%). In vitro protein digestibility of hiPSC-generated proteins was significantly higher (78.0 ± 0.7%) than that of the commercial beef protein isolate (75.7 ± 0.6%). Taken together, this is the first study to report an advanced PG-3DSUSP culture system to produce highly efficient hiPSC-generated proteins that possess more essential amino acids and better digestibility. The hiPSC-generated proteins with superior nutrition quality may be of particular significance as novel alternative proteins in food engineering and industries for future food, beverage, and supplement applications.

## 1. Introduction

As the most important component of animal and human tissues, proteins are large macromolecules consisting of mainly 20 amino acids (AAs) [1]. The body requires complete proteins in adequate quantities for the synthesis of tissue proteins and the maintenance of normal metabolic functions [2]. Generally, AAs can be divided into essential amino acids (EAA) and non-essential amino acids (NEAA). EAA are particularly important nutrients since they cannot be synthesized in humans and therefore must be provided in the diet from dietary protein sources [2,3]. Dietary protein sources mainly come from plant- and animal-based food products. Plant-based proteins have become more popular due to their sustainability and lower production costs [4]. However, plant-based proteins are sometimes incomplete by missing or reducing specific EAAs such as lysine, methionine, and tryptophan [5]. Although animal-based products (e.g., eggs, milk, meat, fish, and shrimp) contain necessary proteins with all EAA [6], the global production of animal proteins is costly and faces sustainability challenges. Animal agriculture usually produces 1 kg of high-quality animal proteins by feeding as much as 6 kg of plant proteins to livestock, which impacts land and water resources as well as greenhouse gas emissions [7]. Additionally, animal-based sources of high EAA, particularly red meat, are associated with the risk of many chronic diseases, such as cancer, cardiovascular diseases, diabetes, and obesity [8].

The Recommended Dietary Allowance (RDA) of proteins for adults, based on minimum physical activity, is 0.8 g/kg body weight per day, but special groups (e.g., infants, children, pregnant, and seniors) need more proteins to meet growth or aging needs [1,9,10]. High-protein diets (defined as exceeding the current RDA) are heavily promoted by athletes as “the gold standard” for building muscle mass and/or losing body fat [11]. Athletes are recommended to consume 1.6–2.4 g/kg each day during weight loss [12]. However, a wide range of ecological issues have decreased the availability of viable agricultural land, freshwater, and fossil fuels [13]. This issue in global food consumption is challenged by human protein needs [14]. Around 800 million people are chronically undernourished due to the challenges that humanity faces in ensuring food security and sustaining the environment [13]. A sustainable solution to food or protein insecurity, especially the provision of high-quality food proteins from alternative resources for humans, appears urgently needed in the face of these nutritional deficiencies [15].

In recent years, a modern biotechnology approach that accelerates food production to achieve the “zero hunger” goals set by the United Nations has been the application of cell culture, marker-assisted selection, and genetic engineering [16]. Cell culture is a widely used in vitro tool used for mechanisms of disease, drug action, tissue engineering, and protein production [17]. Cell culture technology allows cells to grow in bioreactors, reducing land use for agriculture [16]. Over the past few decades, regenerative agriculture has become an emerging research field to produce protein products using muscle stem cells (MSCs) and tissues in cell culture systems [18]. However, there are some challenges with MSCs for protein production, such as maintaining their stem integrity after a few expansion generations and scaling up [19]. Unlike MSCs, pluripotent stem cells (PSCs) or induced pluripotent stem cells (iPSCs) are good candidates for protein regeneration. They can be derived from blood stem cells using a highly efficient method and are indefinitely renewable [20]. Here, the Sun’s lab identified a novel peptide hydrogel (PGmatrix) for three-dimensional (3D) hiPSC maintenance and expansion [21] that would have the potential for cell-based protein production.

For more than a decade, hiPSCs have been cultured in a two-dimensional (2D) monolayer system with specifically designed coating materials in unnatural cell environments [22]. However, hiPSCs in 2D cell culture lack an appropriate stem cell niche, which leads to poor maintenance of pluripotency and unwanted differentiation [21]. Due to the absence of physiological properties in real tissues, 2D cell culture is a simplified and unrealistic condition for cell growth [23]. Considering the limitations of 2D culture methods, the 3D cell culture system mimics the complexity of the extracellular matrix (ECM) and the physiological relevance in vivo [24]. Several 3D cell culture platforms have been developed for hiPSCs, including hydrogels, scaffolds, and decellularized tissues [21]. For example, ECM protein-based hydrogels and RADA peptide hydrogels [25,26], or natural polymers [27,28], have been developed for iPSCs as well as polyethylene glycol (PEG)-based hydrogels (i.e., PNIPAAm-PEG or Mebiol Gel™) [29,30]. However, these 3D methods do not produce high-quality iPSCs compared to the PG-3D system [21,31]. In addition, the PGmatrix-hiSPC can be handled at room temperature or 37 °C and neutral pH condition for cell encapsulation and harvest [21]. Therefore, in this study, the PG-3D system was used to further develop the PGmatrix 3D Suspension (PG-3DSUSP) system to enable a broader range of processes, which is likely to become an increasingly attractive alternative for 2D cell culture [32]. With the PG-3DSUSP system, we improved the scales by 20 times while maintaining the hiPSCs growth performance in terms of cell growth rate, viability, and pluripotency compared to the PG-3D-embedded hydrogel system. The nutritional value of hiPSC-generated proteins is also improved. This study is the first time the nutritional quality of hiPSC-generated protein from PG-3DSUSP, including amino acid composition and protein digestibility, has been explored.

## 2. Materials and Methods

### 2.1. Materials

hiPSCs derived from human fibroblasts were purchased from Applied Stemcell (Milpitas, CA, USA). PGmatrix-3D Suspension (PG-3DSUSP), PG-3D hydrogel, and PGworks were the product of PepGel LLC (Manhattan, KS, USA). hiPSC 3D colony pellets were lyophilized using a FreeZone 6 L Console Freeze Dryer (LabConco, Kansas City, MO, USA). Trypsin (T7409 Trypsin from porcine pancreas Type II-S, lyophilized power, 1000–2000 units/mg dry solid), chymotrypsin (C4129 α-Chymotrypsin from bovine pancreas C4129 Type II, lyophilized power, ≥40 units/mg protein), and protease (P0029 Protease from *Bacillus* sp.) were purchased from Sigma Chemical Co. (St. Louis, MO, USA). Beef protein isolate was purchased from Bulk Supplements (Henderson, NV, USA). Human skeletal muscle protein data was obtained from Gorissen et al. (2018) [4].

### 2.2. hiPSCs 3D Physiological Colony Culture in PGmatrix3D Suspension

hiPSC 3D physiological colony (spheroids) culture was performed following the PGmatrix-3D-Suspension (PG-3DSUSP) using a guide (PepGel LLC, Manhattan, KS, USA). mTeSR1 medium (StemCell Technologies) supplemented with PGgrow (PepGel LLC) at a ratio of 1000:1 (*v*/*v*, mTeSR1: PGgrow) was used to maintain the suspension cell culturing. Briefly, hiPSC cell suspension with a cell density of 1.5–2 × 10^5^ cells/mL was mixed with PG-3DSUSP solution and hydrogelation trigger (PGworks) at ratios of 2:1:0.03 (*v*/*v*) (cell suspension: PG-3DSUSP: PGworks). The mixture with a final cell density of 1–1.5 × 10^5^ cells/mL was then transferred into a 6-well plate (3.94 mL/well) and incubated at 37 °C for cell culture. To feed the cells, 2–3 mL of complete culture medium per well were added to the 6-well plate at days 1, 3, and 4, respectively, and pipetted gently to distribute the fresh medium uniformly into the 3D suspension culture system. Axio Vert A1 miceoscope (Carl Zeiss Microscopy, Munich, Germany) was used to monitor the hiPSC 3D colonies morphology and size evaluation.

### 2.3. hiPSC 3D Colonies Harvesting

hiPSC 3D colonies were harvested on Day 5 of suspension culture, which was also following the PG-3DSUSP using guide (PepGel LLC). Briefly, the 3D suspension culture system was mechanically disrupted thoroughly by pipetting, and then the mixture was transferred to a 50-mL conical centrifuge tube. Then, the mixture was centrifuged at 700× *g* for 5 min by using a swing bucket centrifuge; the supernatant was discarded, and then 3D colony pellets were collected from the tube bottom. Then, the hiPSC 3D colony pellet was lyophilized by Labconco FreeZone 6 L console freeze drying for further characterization.

### 2.4. hiPSCs 3D Colonies Passage

The hiPSC 3D colonies obtained from the 3D culture were dissociated into single or small cluster cells using TrypLE™ Express Enzyme (1X) (Thermal Scientific Fisher, Waltham, MA, USA) at 37 °C for 15–20 min. Then, the cells were passaged following the same 3D culture procedure described above. The cell number and viability were measured using acridine orange/propidium iodide (AO/PI) assay from Nexcelom Bioscience and counted using a Cellometer Auto 2000 (Nexcelom Bioscience LLC, Lawrence, MA, USA).

### 2.5. hiPSC 3D Physiological Colony Culture in PGmatrix-hiPSC Culture

hiPSC 3D colonies cultured in PGmatrix-hiPSC (PG-3D) hydrogel were used as the control for comparison purposes following the PG-3D user guide (PepGel LLC, Manhattan, KS, USA). The hiPSC seeding density of the cells was 2 × 10^5^/mL, and the cells were embedded in a 0.5 mL gel volume using a 24-well plate. The hiPSC 3D colonies were harvested following the PGmatrix-hiPSC user guide (PepGel LLC, Manhattan, KS, USA). To ensure full nutrient penetration into the hydrogel, the thickness of the hydrogel was set to less than or equal to 2 mm.

### 2.6. Real-Time Quantitative PCR (RT-qPCR)

Total RNA samples were extracted from each cell sample using the Direct-zol RNA MiniPrep kit (Zymo Research Corp., Irvine, CA, USA) and diluted to 10 ng/µL. RT-qPCR reactions were conducted with the Bio-Rad CFX96™ Touch™ Real-time PCR Detection System using an iTaq Universal SYBR Green One-Step Kit (Bio-Rad, Hercules, CA, USA). The first reverse transcription reaction was set at 50 °C for 10 min, followed by polymerase activation and cDNA denaturation at 95 °C for 1 min. The reactions then continued with 45 cycles of denaturation at 95 °C for 10 s and annealing and extension at 60 °C for 40 s for hTERT and 3 housekeeping genes (*hEID2*, *ZNF324B*, and *hCAPN10*) [33] or annealing at 52 °C for 30 s and extension at 72 °C for 20 s for the other 4 target genes (*UTF1*, *NANOG*, *OCT4*, and *REX1*). The sequences of all primers are listed in Table 1. Each gene in all samples was assayed in triplicate. The Ct values were analyzed using Bio-Rad CFX Manager 3.0 software. The expression fold change in a target gene in a tested sample was compared with that of a control sample and normalized to the average expression levels of three housekeeping genes.

### 2.7. Amino Acid Profile Analysis of hiPSC 3D Colonies

Approximately 3 mg of hiPSCs were hydrolyzed with 6 M HCl for 24 h at 110 °C. Then, the resulting amino acids were quantified with hydrophilic interaction chromatography coupled tandem mass spectrometry (HILIC-MS/MS) using a Shimadzu Nexera X2 UHPLC system connected to a SCIEX QTRAP 6500+ triple quadrupole-linear ion trap mass spectrometer, equipped with an IonDrive™ Turbo V electrospray ionization source, as described previously [34]. Positive ion mode was used for all amino acids except cysteic acid, which was analyzed in negative mode. The sample was injected into an Infinity Lab Poroshell 120 Z-HILIC column (2.7 μm, 100 × 2.1 mm; Agilent Technologies, Santa Clara, CA, USA), and amino acids were eluted with a gradient of ammonium formate in water (A) and acetonitrile:water (90:10), pH 3.0, at a final concentration of 20 mM ammonium formate (B) with a constant flow of 0.25 mL/min, followed by 50% B over 6 min, then 100% B over 30 s, followed by 6.5 min to re-equilibrate the column. The electrospray ionization source parameters were as follows: ion spray voltage: 4.5 kV (ESI+ and ESI−); ion source temperature: 400 °C; source gas: 1:45; source gas: 2:40; and curtain gas: 35.

### 2.8. In Vitro Protein Digestion

The pH-drop procedure of Hsu et al. (1997) [35] was adopted and applied in this study. Briefly, 62.5 mg of hiPSC proteins were dispersed in 10 mL of distilled water at 37 °C for 1 h. Ten milliliters (10 mL) of a multi-enzyme solution were prepared, containing 16 mg of trypsin (T7409 Trypsin from porcine pancreas Type II-S, lyophilized power, 1000–2000 units/mg dry solid), 31 mg of chymotrypsin (C4129 α-Chymotrypsin from bovine pancreas C4129 Type II, lyophilized power, ≥40 units/mg protein), and 13 mg of protease (P0029 Protease from *Bacillus* sp.) Protease from *Bacillus* sp. was used to replace the discontinued peptidase. The multi-enzyme solution was prepared fresh on the day of analysis and kept at 37 °C, and its pH was adjusted to about 8.0 as described above [36].

Upon rehydration, 1 mL of the multi-enzyme solution was added to the 10 mL protein solution. The pH variation was recorded after a 10-min reaction. The pH at 10 min of digestion (ΔpH_10min_) was used to estimate protein digestibility using the equation below:Y = 65.66 + 18.10ΔpH_10min_.
where ΔpH_10min_ = pH_initial_ − pH_final_, pH_initial_ is the pH after stabilizing approximately 8.0, and pH_final_ is the pH of the solution 10 min after the enzyme reaction.

### 2.9. Statistical Analysis

The protein digestibility results were analyzed in triplicate and expressed as means ± SD. The results were used in Student’s *t*-test to compare results (*p* ≤ 0.05). A statistical analysis was conducted using the JASP statistical system, version 0.16.3 (JASP Team, 2022) [37].

## 3. Results and Discussion

### 3.1. The hiPSC Production System

Peptide-based PGmatrix 3D hydrogel (PGmatrix-hiPSC (PG-3D)) was determined to be the superior 3D culture for generating physiological hiPSC 3D colonies in a previous study [21], but with limitation of scalable production due to operability and economic considerations. Alternatively, a novel peptide-based PGmatrix-3D suspension culture (PG-3DSUSP) system was identified for cultured hiPSC protein production. Figure 1 presents the scheme diagram of the PG-3D versus PG-3DSUSP culture workflow. hiPSC cells were suspended in PG-3DSUSP systems at day 0, and fresh medium was added to the suspension culture by mixing the fresh medium with the PG-3DSUSP culture system to feed cells on day 1, 3, and 4 to reach 10 mL volume. By day 5, cell spheroids were harvested by directly centrifuging the PG-3DSUSP culture system. The cell spheroid pellet was collected from the bottom of the centrifuge tube for further analysis. Compared with the PG-3D system, PG-3DSUSP culture is easily operable and the nutrients penetration process was omitted by mixing culture medium with the hydrogel. In addition, cultured medium containing biologics released from cells during culture can easily be recovered from the supernatant on day 5 in the cell spheroids harvesting by centrifuge step.

hiPSCs growth within PG-3DSUSP: A single hiPSC cell or small clusters cultured in the PG-3DSUSP system at a seeding density of 1 × 10^5^ cell/mL at day 0. Cell expansion in a 3D manner was observed; the developed hiPSC 3D colonies had a dimeter range of 20–30 μm by day 3 of culturing, compared to the cell sizes of day 0 with 10 μm (Figure 2A,B). At day 5, the size of hiPSC 3D colonies reached 30–60 μm (Figure 2C). TrypLE was used to trypsinize the 3D colonies into single or small cluster cells for proliferation and viability measurements. The results showed that with total seeding cells of 2.4 × 10^6^ per 6-well plate (4 × 10^5^ per well), cells proliferated to 6–8 × 10^7^ of hiPSCs (~5–7 × 10^5^ 3D colonies) by day 5. Dry matter of crude protein was about 2.5–3.0 mg per 10 mL volume or per well of the 6-well plate. The overall cell expansion fold across multiple passages was well maintained in the range of 15–20 folds with viability of 95–99% in 5 days. The PG-3DSUSP culture system presents comparable cell growth performance to PG-3D culture in terms of cell morphology and growth rate. Both 3D culture systems generated hiPSC 3D colonies with sizes of 30–50 μm (Figure 2D), and with a comparable cell proliferation of 15–20 folds and viability above 95% (Figure 3). The growth performance of hiPSC within PG-3D is in agreement with the previous study [21].

Besides its stable long-term cell maintenance, PG-3D hydrogel also proved to generate hiPSC 3D colonies (spheroids) with superior pluripotency compared with that from traditional 2D culture or other Matrigel, Vitronectin, or existing Polyethylene glycol (PEG)-based 3D cell culture method. Several gene markers were selected in this study to characterize the stemness of hiPSCs from PG-3DSUSP. Table 2 presents the fold changes in gene expression that were normalized using PG-3D-cultured 3D colonies as a reference. hiPSC 3D colonies grown in PG-3DSUSP have similar or higher expression levels of NANOG (1.46), OCT4 (1.99), hTERT (1.55), REX1 (0.88), and UTF1 (1.09) than control (1), indicating that the pluripotency of hiPSCs can be maintained equally or better in PG-3DSUSP than in PG-3D hydrogel. Overall, the PG-3DSUSP culture system proved to be a competitive culture system compared to the well-established PG-3D hydrogel, but with the distinctive advantages of an easily culturing workflow and great potential for scalable cell production.

Some studies intended to culture hiPSCs in 2D culture systems with specifically designed coating materials, resulting in flat and stretched morphologies of hiPSCs [21]. However, a 2D cell culture system does not mimic the environment of the human body, which affects cell processes in hiPSCs such as proliferation, apoptosis, differentiation, gene expression, and drug sensitivities [38]. Moreover, 2D cell-cultured cells undergo cytoskeletal rearrangements and acquire artificial polarity, resulting in aberrant gene and protein expression [39].

Hydrogels (natural- and synthetic-based) have been considered to provide an extracellular matrix (ECM)-like scaffold for various stem cells due to their 3D nature and high water content, which may potentially replace 2D cell culture [32,40]. We have tried to provide more realistic biochemical and biomechanical microenvironments for hiPSCs. It is reported that high-efficient physiological formation of hiPSC spheroids with stable genetic integrity was developed within PGmatrix-hiPSC 3D hydrogel (PG-3D) [21]. PG-3DSUSP is a novel technology for suspension culture, specifically designed for scaling up cell manufacturing. Under the PG-3DSUSP setting, cell growth performance remained consistent through multiple passages, with cell increases of 15–20 fold and cell viability of 95–99%. In addition, hiPSCs from PG-3DSUSP showed similar or higher stemness gene expression levels in comparison to those from PG-3D. Overall, PG-3DSUSP presents similar growth performance in terms of growth rate and gene integrity compared with the well-established PG-3D hydrogel system. The superior advantages of PG-3DSUSP are its easy operability, cost effectiveness, and potential for large-scale cell culturing and protein production from cultured cells.

### 3.2. hiPSC Nutritional Properties

The amino acid composition of hiPSCs versus human skeletal muscle protein is shown in Figure 4. The total essential amino acid content of hiPSCs (39.0%) was significantly higher than that of human skeletal muscle protein (31.8%) (Figure 4A). Among those, the content of isoleucine (4.7%), leucine (8.8%), lysine (8.5%), phenylalanine (4.7%), threonine (3.9%), and valine (5.0%) of hiPSCs was higher than human skeletal muscle proteins, and the content of histidine (2.3%) and methionine (1.1%) was lower than human skeletal muscle proteins. The total non-essential amino acid content of hiPSCs (46.1%) was also significantly higher than human skeletal muscle protein (29.0%) (Figure 4B). In vitro protein digestibility values of hiPSCs and commercial beef protein isolates are shown in Figure 5. The in vitro protein digestibility of hiPSCs (78.0 ± 0.7%) was significantly improved compared with commercial beef protein isolates (75.7 ± 0.6%, *p* < 0.015).

The determination of amino acid composition and in vitro protein digestibility revealed the improved nutritional value of hiPSC-generated proteins. The total EAA of hiPSCs (39.0%) was higher than that of human skeletal muscle protein (31.8%). We chose human skeletal muscle protein for comparison with hiPSCs because it is reasonable that they both come from humans. Additionally, when focusing on muscle protein synthesis, we included human skeletal muscle protein as a reference protein with an “ideal” amino acid composition [4]. Leucine content (8.8%), potentially a key mechanism translating diet quality into the muscle protein synthesis response to meals [4,41], was the highest in EAA. It has been shown that leucine stimulates Sestrin2 to translocate to the lysosomal membrane, which activates mTORC1 and results in muscle protein synthesis [42]. Hence, leucine contents are an important factor in modulating muscle protein synthesis after protein ingestion [4]. The content of lysine (8.4%) was also higher than human skeletal muscle (6.6%). Lysine and methionine are necessary amino acids for making carnitine, which plays a vital role in the metabolism of fatty acids and energy production. However, histidine content (2.3%) and methionine content (1.1%) were lower than human skeletal muscle protein. Histidine content of hiPSCs reached the WHO/FAO/UNU requirements [43].

The in vitro protein digestibility of hiPSCs (78.0 ± 0.7%) was higher than commercial beef protein isolates (75.7 ± 0.6%). Studies have shown that animal-based proteins digest more easily than plant proteins [44]. Compared with other animal-based sources, beef protein is a high-quality source that is commonly used on the market. In this study, we compared the protein digestibility of hiPSCs with commercial beef protein isolates as the control group. A high correlation was found between the pH of a protein suspension immediately after 10 min of digestion in the multienzyme solution and in vivo apparent digestibility [35], particularly when the protein sources were analyzed by plant or animal origin [45]. Additionally, as in vivo methods are expensive, time-consuming, and involve ethical issues, they are not suited to studying multiple samples required to understand ingredient interactions or processing effects, which are commonly assessed using in vitro techniques [46]. The pH drop three-enzyme (trypsin-chymotrypsin-peptidase) method is widely used to estimate in vitro protein digestibility that determines the pH after 10 min of reaction [35]. Protease was used to replace the discontinued peptidase to build the pH vs. digestibility calibration [36]. This was based on the principle that hydrolysis by amino acid carboxyl groups deionizes and releases free protons [47]. This in vitro method using the multienzyme system was highly correlated with in vivo apparent digestibility [35]. It improved the prediction of in vivo protein digestibility, was reproducible, and predicted in vivo (rat fecal) digestibility accurately [45]. hiPSC-generated protein showed higher protein digestibility than animal protein. Intestinal protein availability can be estimated from protein digestibility, which reflects the efficiency of protein utilization in the diet [48]. It is well known that the protein digestibility of animal protein is higher than that of plant protein [49]. There is a possibility that infants or the elderly may have insufficient digestion of animal proteins due to low secretion of digestive juices or enzymes in the immature state or a malfunctioning GI tract [50]. hiPSCs have higher in vitro protein digestibility than animal proteins, which can possibly be applied to food products for infants and the elderly in the future. Clearly, in vitro methods do not mimic the complexity of human digestion. In the future, it will be necessary to investigate in animal models or in vivo whether hiPSC-generated protein could be edible or have any potential side effects.

This study demonstrated that culturing hiPSCs in an advanced 3D peptide hydrogel (PG-3DSUSP) system is an applicable method for lab-scale cell-cultured protein production. The cultured hiPSCs exhibited improved nutritional attributes in terms of amino acid composition and in vitro protein digestibility for the first time.

## 4. Conclusions

A 10 mL lab-scale PG-3DSUSP hydrogel presents similar hiPSC growth performance to the well-established PG-3D hydrogel culture system, with a consistent hiPSC growth fold expansion of 15–20 times in 5 days and viability of 95–99%. The pluripotency gene expression levels of hiPSC from PG-3DSUSP were like those from PG-3D hydrogel. In short, PG-3DSUSP is the superior 3D suspension culture hydrogel with the potential for distinctive advantages for large-scale cell manufacturing. hiPSC-generated proteins from the PG-3DSUSP culture system contain higher levels of total essential amino acids (EAA) and improved digestibility. A high abundance of leucine in hiPSC-generated proteins is particularly useful since it is a key factor translating diet quality into the muscle protein synthesis response to meals. The highly efficient production of hiPSC-generated protein using advanced 3D peptide hydrogel with more essential amino acids and better digestibility appears of particular significance as a novel alternative protein in food engineering and industries for future food, beverage, and supplement applications.

## Figures and Tables

**Figure 1 foods-12-02713-f001:**
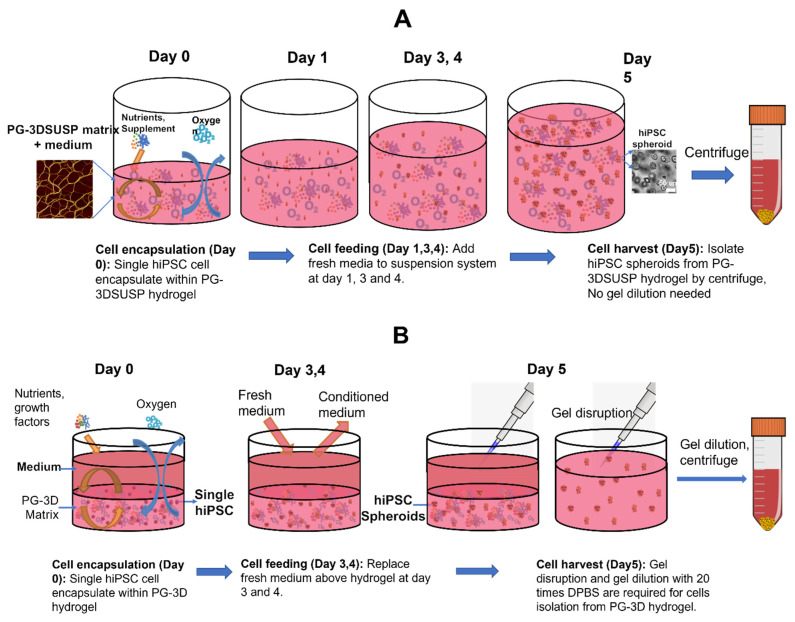
(**A**) Schematic diagram of PGmatrix-3D Suspension (PG-3DSUSP) culture workflow using a well plate. (**B**) Schematic diagram of PGmatrix-hiPSC 3D (PG-3D) culture workflow using a well plate.

**Figure 2 foods-12-02713-f002:**
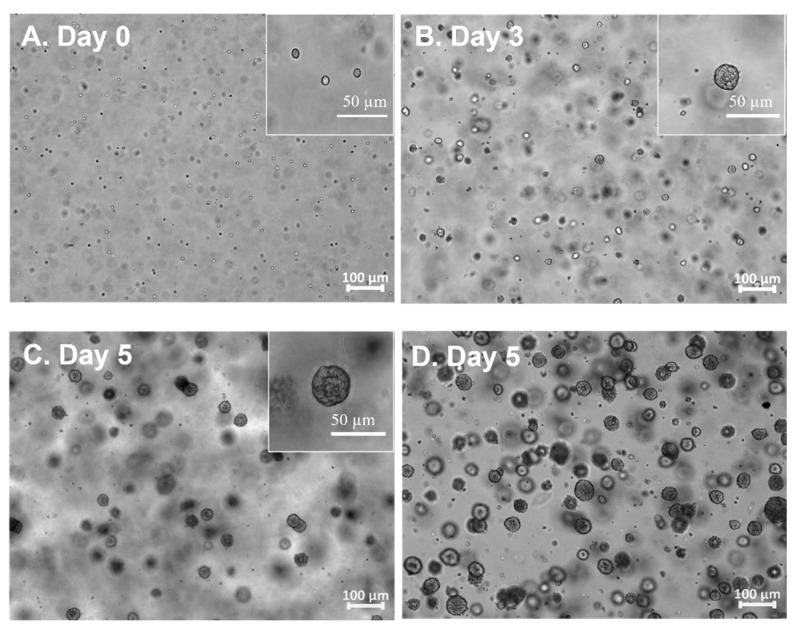
Morphology of hiPSC grown in PGmatrix-3D-Suspension (PG-3DSUSP) system from day 0 to day 5 (**A**–**C**); hiPSC grown in PGmatrix-hiPSC (PG-3D) hydrogel at day 5 (**D**).

**Figure 3 foods-12-02713-f003:**
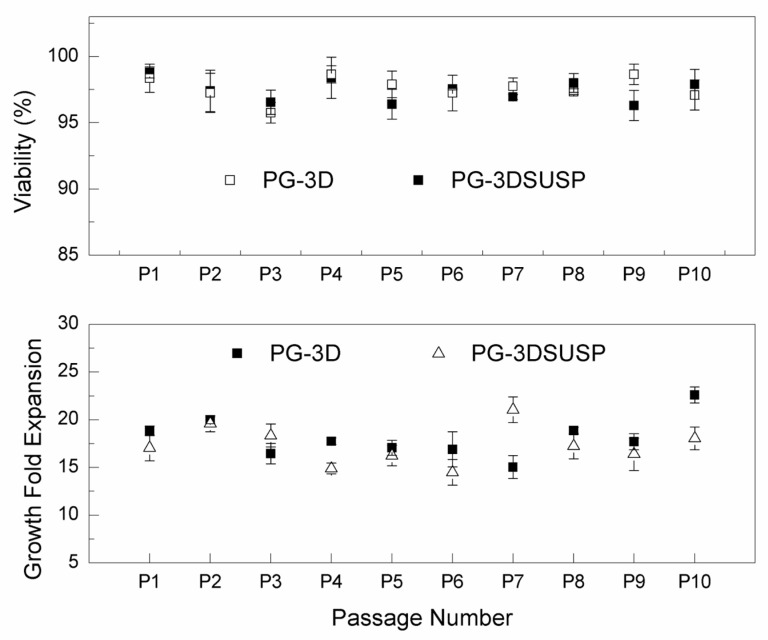
The hiPSC growth fold expansion and viability in PG-3DSUSP and PGmatrix-hiPSC (PG-3D) hydrogel on day 5.

**Figure 4 foods-12-02713-f004:**
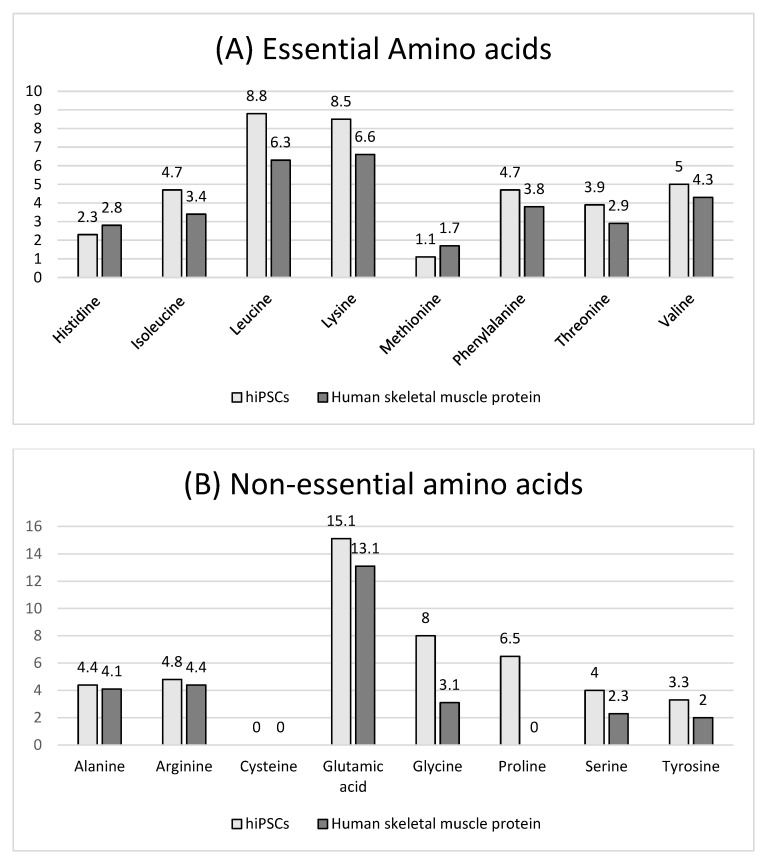
Amino acid composition (mol %) of hiPSCs and human skeletal muscle protein (sources from Gorissen et al., 2018 [4]). (**A**) Essential amino acids, (**B**) Non-essential amino acids.

**Figure 5 foods-12-02713-f005:**
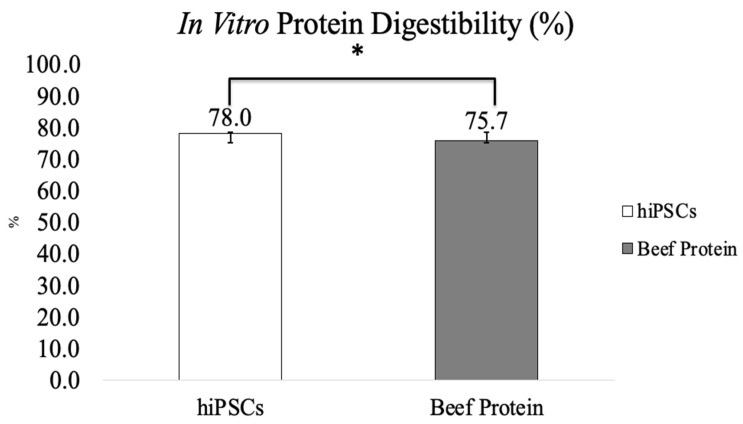
In vitro protein digestibility of hiPSCs and commercial beef protein isolate (Results are shown as mean ± SEM, *: *p* value < 0.05).

**Table 1 foods-12-02713-t001:** Primers of five target genes (*hTERT*, *NANOG*, *OCT4*, *REX1*, and *UTF1*) and three house keep genes (*hEID2*, *ZENF324B*, and *CAPN10*) used for RT-qPCR analysis.

Gene	Sequence
*hTERT*	Forward (5′ to 3′): GGAGCAAGTTGCAAAGCATTG
	Reverse (3′ to 5′): TCCCACGACGTAGTCCATGTT
*NANOG*	Forward (5′ to 3′): TGTGATTTGTGGGCCTGA
	Reverse (3′ to 5′): GTGGGTTGTTTGCCTTTG
*OCT4*	Forward (5′ to 3′): AAAGAGAAAGCGAACCAG
	Reverse (3′ to 5′): CCACATCCTTCTCGAGCC
*REX1*	Forward (5′ to 3′): GTTTCGTGTGTCCCTTTC
	Reverse (3′ to 5′): CTTTCCCTCTTGTTCATTC
*UTF1*	Forward (5′ to 3′): CTCCCAGCGAACCAG
	Reverse (3′ to 5′): GCGTCCGCAGACTTC
*hEID2*	Forward (5′ to 3′): GAAGCCTGCAGAGCAAGG
	Reverse (3′ to 5′): ATATCGAGGTCCACCCTGTG
*ZENF324B*	Forward (5′ to 3′): GAGAATGGCCACGAGCTTT
	Reverse (3′ to 5′): TTTACACTGTGGCAGGCATC
*hCAPN10*	Forward (5′ to 3′): GGAGGTGACCACAGATGACC
	Reverse (3′ to 5′): GTAAGGGGAGCCAGAACACA

**Table 2 foods-12-02713-t002:** RT-qPCR analysis of five hiPSC pluripotency-related gene expressions from PGmatrix-3D Suspension (PG-3DSUSP) culture system versus PGmatrix-hiPSC (PG-3D) culture system.

8	hiPSC Pluripotency-Related Gene Expression				
	*REX1*	*OCT4*	*NANOG*	*UTF1*	*hTERT*
PG-3D	1	1	1	1	1
PG-3DSUSP	0.88	1.99	1.46	1.09	1.55

## Data Availability

The datasets generated for this study are available on request to the corresponding author.

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
