# Peer review of "High Nutritional Quality of Human-Induced Pluripotent Stem Cell-Generated Proteins through an Advanced Scalable Peptide Hydrogel 3D Suspension System"

_foods, 2023, doi:10.3390/foods12142713_

Round 1

Reviewer 1 Report

The authors should reconsider the use of scalable in the text because 10ml is not really a scalable process yet. It is not clear if the gel could be used as beads in suspension since 3D hydrogels have significant limitations for using cell culture to produce protein. The authors should clarify how much (mass of protein) can be produced in 10ml and comment the possible advantages or challenges. Should be clarified if there are other biomolecules produced with the protein. 

Author Response

Thanks for the comments, in fact, the 10-ml scale was conceptualized for scalable process as the 0.5-ml scale was for hydroggel embedded 3D culture. The main challenge issue for gel embedded culture is the penetration of nutrients and air. For the 0.5-ml culture scale, nutrients can only penetrate maximum 2 mm in thickness, which limits its scaling up. While for the 10-ml scale, the hydrogel can be formulated to allow nutrients penetrate the hydrogel without limitation. Larger scales such as 500 ml up to 5 L are under investigation. There are several scaling up parameters that need to be optimized during scaling up and that are related to cell performance or protein production. These scaling up studies are not intended in this current paper, which will be another well-defined work.

Additional comments: The authors should reconsider the use of scalable in the text because 10ml is not really a scalable process yet. It is not clear if the gel could be used as beads in suspension since 3D hydrogels have significant limitations for using cell culture to produce protein. The authors should clarify how much (mass of protein) can be produced in 10ml and comment the possible advantages or challenges. Should be clarified if there are other biomolecules produced with the protein.

               Response: Per suggested, the scalable wording has been deleted or reworded throughout the whole manuscript.

               The cell culture using 3D hydrogels in suspension has been described in detail in sections 2.2 for cell culture, section 2.3 for cell harvesting, and section 3.1 for the biologicals released from cells.

     Dry matter of crude protein was about 2.5-3.0 mg per 10 mL volume or per well of 6-well plate,          which has been clarified in lines 284-285. 

Reviewer 2 Report

In this article the authors present their new advanced scalable hydrogel 3D suspension system, named PGMatrix 3D suspension (PG-3DSUSP), whose properties are analyzed in comparison with their previous PGmatrix peptide hydrogel for 3D embedded culture of long term hiPSCs maintenance and expansion (PGmatrix-hiPSC (PG-3D)). These hydrogels are already commercialized on the PepGel website, as an example here: https://pepgel.com/product/pgmatrix-3d-suspension-hipsc-kit

PepGel is a company spun-out of Kansas State University to exploit the hydrogel platform’s technology probably by some of the authors of the actual article. I suppose this was made possible due to Xiuzhi Susan Sun which also have the patents US11135163B2 and EP2928513B1 in the field.

PG-3DSUSP proved to be a competitive culture system versus the older and well-established PG-3D hydrogel having the advantages of an easily culturing workflow and great potential of scalable cell production as it needs only centrifugation and no gel dilution is needed as is the case for PG-3D hydrogel.

The presentation of their hydrogel is relatively well done and in my opinion the article can be published with minor corrections in the special issue “Advances in Novel Foods, Gut Microbiota, and Human Health” of Food Nutrition journal.

I only identified some small English mistakes:

Line 332: “animal-baed proteins” must be written “animal-based proteins”

Line 344: “Protease used to replace” intent was to say “Protease was used to replace” but in fact all decription of the pH drop three-enzyme method is not written clearly enough and there are also mistakes as you will see below

Line 345: “amnio acid” must be written “amino acid”

Line 355: “which applied for food products in infants” can be expressed better as “which can possibly be applied for foods in infants”

It will be better that the authors reread with care the article one more time.

The English is relatively good I identified only some minor mistakes.

Author Response

Comment 1: line 332: “animal-baed proteins” must be written “animal-based proteins”.

Response: Per requested, line 332: “animal-baed proteins” has been changed to “animal-based proteins”

Comment 2: Line 344: “Protease used to replace” intent was to say “Protease was used to replace” but in fact all decription of the pH drop three-enzyme method is not written clearly enough and there are also mistakes as you will see below

Response: Per requested, line 344: “Protease used to replace” has been changed to “Protease was used to replace”         

Comment 3: Line 345: “amnio acid” must be written “amino acid”

Response: Per requested, Line 345: “amnio acid” has been changed to “amino

Comment 4: Line 355: “which applied for food products in infants” can be expressed better as “which can possibly be applied for foods in infants”

Response: Per requested, Line 355: “which applied for food products in infants” has been changed to “which can possibly be applied for foods in infants”      

Comment 5: It will be better that the authors reread with care the article one more time.

Response: Per requested, the whole manuscript has been reread twice.